# Management of High-Risk Pulmonary Embolism: What Is the Place of Extracorporeal Membrane Oxygenation?

**DOI:** 10.3390/jcm11164734

**Published:** 2022-08-13

**Authors:** Benjamin Assouline, Marie Assouline-Reinmann, Raphaël Giraud, David Levy, Ouriel Saura, Karim Bendjelid, Alain Combes, Matthieu Schmidt

**Affiliations:** 1Médecine Intensive Réanimation, Institut de Cardiologie, Assistance Publique-Hôpitaux de Paris, 75013 Paris, France; 2Cardiology Department, AP-HP, Sorbonne Université, Pitié-Salpêtrière University Hospital, 75013 Paris, France; 3Intensive Care Unit, Geneva University Hospitals, 1205 Geneva, Switzerland; 4Faculty of Medicine, University of Geneva, 1205 Geneva, Switzerland; 5Sorbonne Université, GRC 30, RESPIRE, UMRS 1166, ICAN Institute of Cardiometabolism and Nutrition, 75013 Paris, France

**Keywords:** high-risk pulmonary embolism, massive pulmonary embolism, VA-ECMO, reperfusion therapy, ECPR

## Abstract

Pulmonary embolism (PE) is a common disease with an annual incidence rate ranging from 39–115 per 100,000 inhabitants. It is one of the leading causes of cardiovascular mortality in the USA and Europe. While the clinical presentation and severity may vary, it is a life-threatening condition in its most severe form, defined as high-risk or massive PE. Therapeutic options in high-risk PE are limited. Current guidelines recommend the use of systemic thrombolytic therapy as first-line therapy (Level Ib). However, this treatment has important drawbacks including bleeding complications, limited efficacy in patients with recurrent PE or cardiac arrest, and formal contraindications. In this context, the use of venoarterial extracorporeal membrane oxygenation (VA-ECMO) in the management of high-risk PE has increased worldwide in the last decade. Strategies, including VA-ECMO as a stand-alone therapy or as a bridge to alternative reperfusion therapies, are associated with acceptable outcomes, especially if implemented before cardiac arrest. Nonetheless, the level of evidence supporting ECMO and alternative reperfusion therapies is low. The optimal management of high-risk PE patients will remain controversial until the realization of a prospective randomized trial comparing those cited strategies to systemic thrombolysis.

## 1. Introduction

Pulmonary embolism (PE) is a common disease with an annual incidence rate ranging from 39–115 per 100,000 population [1,2] and it is one of the leading causes of cardiovascular mortality in the United States with 300,000 deaths per year [1]. The clinical presentation and severity may vary widely. Its most severe form, defined as high-risk or massive PE, has an incidence rate of 5% and is associated with a 30-day mortality rate ranging from 16 to 46% for patients in shock and approaching 52 to 84% for those with cardiac arrest [3,4].

Therapeutic options in high-risk PE are limited. Current European guidelines recommend the use of systemic thrombolytic therapy as the first-line therapy (Level Ib) [5]. However, this treatment has important drawbacks including bleeding complications and limited efficacy in patients with recurrent PE or cardiac arrest [6]. Besides, a significant percentage of the patients have formal and relative contraindications to this therapy [7]. Other treatment options, including surgical embolectomy, have been associated historically with poor outcomes [8]. The use of extracorporeal life support (ECLS) as a stand-alone strategy has been advocated by several authors but remains controversial [7,9,10,11]. As per the latest ESC guidelines published in 2019, the use of extracorporeal membrane oxygenation (ECMO) “may be considered in combination with surgical embolectomy or catheter-directed treatment, in refractory circulatory collapse or cardiac arrest” (Level IIB) [5]. To date, there is no randomized controlled trial (RCT) that addresses the place of ECMO, with or without other reperfusion therapies, in the management of high-risk PE. This narrative review aims to summarize the rationale and evidence for ECLS and adjunct therapies in high-risk PE and to discuss the research agenda in this field.

## 2. Pathophysiology

High-risk PE is defined as acute PE with hemodynamic instability, characterized either as persistent hypotension (systolic blood pressure less than 90 mmHg for more than 15 min without signs of organ hypoperfusion), shock (systolic blood pressure less than 90 mmHg with signs of organ hypoperfusion), or cardiac arrest [5]. This condition is characterized by an obstructive shock, induced by an abrupt increase in the pulmonary vascular resistance (PVR) leading to right ventricular (RV) failure. The increase in RV afterload is due to both direct and indirect mechanisms [12]. To significantly increase the pulmonary artery pressure, blood clots must occlude more than 30–50% of the total cross-sectional area of the pulmonary bed [13]. In addition, the release of serotonin and thromboxane A2 induces vasoconstriction and contributes to raising the PVR [14]. As the RV is not conditioned to work against high impedance [15], the RV stroke volume decreases linearly with the afterload elevation [13,16]. This results in RV dilation, which impedes the RV systolic function for several reasons. First, the increased RV end-diastolic volume damages the contractile properties of the myocardium. Second, the RV dilation alters ventricular interdependence with a systolic–diastolic leftward shift of the interventricular septum, which impedes the left ventricle filling and ejection [17,18]. The left ventricular ejection fraction is therefore decreased and systemic hypotension ensues [19,20]. The RV coronary perfusion decreases during both systole (systemic hypotension) and diastole (RV dilation) and causes myocardial ischemia in the context of an imbalance between RV oxygen delivery and demand. Lastly, autopsy findings have reported significant inflammatory infiltrates in the RV myocardium, composed mainly of mononuclear cells and neutrophils granulocytes [21]. This vicious cycle, also called the RV death-spiral, is responsible for the circulatory collapse seen in massive PE.

### 2.1. Systemic Thrombolysis: What Is the Evidence?

***Which patients?*** Thrombolysis is an established treatment in patients presenting an obstructive shock. It has been used in severe PE for more than 50 years. According to the 2019 guidelines of the European Society of Cardiology, systemic thrombolysis is recommended as the first-line therapy in high-risk PE (Level I) [5]. Thrombolysis induces clot dissolution by converting plasminogen into plasmin and decreases the clot burden faster than heparin [22,23] to rapidly improve the RV function, and pulmonary and systemic hemodynamics [22].

The first RCT on systemic thrombolysis (the “urokinase pulmonary embolism trial”) compared urokinase versus heparin therapy alone and included 114 patients [22]. Authors reported significant differences between groups, in favor of the urokinase arm with an unequivocal clot resolution compared to heparin. Although the urokinase regimen did not achieve complete thrombolysis in most cases, thrombolysis can induce a hemodynamic improvement even with incomplete clot resolution. However, the therapeutic window of this treatment is small and 45% of the included patients had hemorrhagic complications. In 1995, Jerjes-Sanchez et al. published the first and only RCT on thrombolysis in high-risk PE [23]. Eight patients were randomized to receive either intravenous streptokinase (1′500′000 IU of streptokinase in 1 h) followed by a heparin infusion or heparin alone. Included patients had similar baseline characteristics, were in cardiogenic shock, and had echographic signs of RV failure. The results were clear. The mortality rate in the streptokinase group was 0% and 100% in the heparin group (*p* = 0.02) leading to the trial being stopped early. Besides, patients randomized in the thrombolysis arm improved their clinical and echocardiographic findings in the first hour after treatment. It is estimated that 35% of the patients died within the first few hours of presentation, stressing that patients with unstable PE need an effective treatment that rapidly restores the hemodynamics. Based on those two studies, systemic thrombolysis became the standard of care in high-risk PE patients even though no confirmatory trials were performed in this population. In the context of intermediate-risk PE, several RCTs and meta-analyses [24,25,26,27,28] have reported lower mortality in the thrombolytic group but with a significant increase in major bleeding.

***Dose optimization?*** As the bleeding complications could be a dose-dependent side effect of thrombolysis, one could ask if a reduction in dose would be as effective and cause less harm? The MOPETT trial [28] and the study of Wang et al. [29] both reported that a low-dose thrombolysis regimen produced similar improvements in RV functions, lung perfusion defects, and pulmonary artery obstructions with a trend toward a lower incidence of bleeding with low-dose thrombolysis regimen events [29], and an unclear effect on mortality [28].

However, both trials were underpowered to detect any difference in mortality. Besides, the incidence of bleeding events and mortality was much lower than previously reported even though more than a third of the patients had a massive PE [29]. Lastly, patients included in both trials had a lower risk mortality profile than those included in historical trials on thrombolysis.

***Does timing matter?*** Thrombolysis timing seems to be of utmost importance, with the greatest probability of success when administered within the first 48 h of the embolic event. This was reported for the first time in the “urokinase pulmonary embolism trial” [22] and later confirmed in an analysis that gathered results from five multicenter trials of thrombolytic therapy for PE [30]. Among 308 patients, there was an inverse association between duration of symptoms and improvement in pulmonary vascular reperfusion after thrombolysis. Similarly, among 488 patients with PE who underwent thrombolysis, 8.2% of them had persistent shock and echocardiographic RV dysfunction [6]. The authors reported an inverse association between duration of symptoms and the probability of thrombolysis success. In addition, 80% of the deaths observed in the repeat-thrombolysis group were reported in patients with longer symptom duration.

### 2.2. Alternative Reperfusion Therapy

The latest guidelines of the European Society of Cardiology recommend the use of alternative reperfusion therapy in high-risk PE patients in “whom thrombolysis is contraindicated or has failed”. Indeed, surgical embolectomy is “recommended,” in this context (class of recommendation I, level C), whereas catheter-delivery thrombolysis “should be considered” (class of recommendation IIa, level C) [5].

***Surgical embolectomy*** is a high-risk procedure, that requires cardio-pulmonary bypass (CBP) and sternotomy. In the context of RV failure and cardiogenic shock, induction of general anesthesia and initiation of mechanical ventilation could promptly lead to hemodynamic deterioration and cardiac arrest before the start of CBP. Furthermore, performing cardiac surgery after a failed thrombolysis exposes patients to potential major intraoperative bleeding and massive transfusion. However, surgical embolectomy could be associated with an improved survival rate according to several recent retrospective studies [10,31,32,33]. While some authors argue that it should be considered as first-line therapy in high-risk PE [34], there is no RCT comparing this strategy to systemic thrombolysis. Furthermore, while systemic thrombolysis is widely available, surgical embolectomy requires a high level of surgical expertise and is not offered in all centers. Besides, the risk of chronic thromboembolic pulmonary hypertension does not seem to decrease with that procedure [11].

***Catheter-delivery thrombolysis*** involves the insertion of a catheter in the pulmonary artery and includes several different techniques to achieve reperfusion. Lysis of the clot can be performed either by mechanical fragmentation, thrombo-aspiration, or pharmacologically, with the delivery of a small dose of thrombolysis. The rationale behind catheter-delivered lysis is to achieve similar effectiveness compared to systemic treatment, but with a lower risk of bleeding complications [24]. Data supporting this strategy are scarce and come from small sample size observational studies [35,36]. In addition, this strategy has never been compared to systematic thrombolysis in an RCT.

### 2.3. Extra Corporal Life Support in High-Risk PE

As stated before, a number of patients have absolute contraindications to thrombolysis and about 8% of patients receiving thrombolysis will not respond to this treatment [6]. While the 2019 ESC guidelines advocate for the use of alternative reperfusion therapy after failed thrombolysis [5], some patients might be too unstable to undergo these procedures. In this context, circulatory support provided by venoarterial (VA)-ECMO could be an efficient salvage therapy. In the modern era of critical care medicine, ECLS is widely available through the ECMO mobile team. In addition, this technique is reliable and can be performed at the bedside, allowing the bridging of refractory patients to several therapeutic options.

***Physiological rational***. VA-ECMO seems to be the perfect way to “break” the so-called “RV death spiral”. First, VA-ECMO is one of the quickest ways to promptly restore hemodynamic stability and provide adequate gas exchange. Second, VA-ECMO is very efficient in supporting RV failure. RV is therefore unloaded by the admission cannula, with a subsequent decrease in RV end-diastolic volume, RV end-diastolic pressure, and RV myocardial oxygen consumption. The balance between RV oxygen demand and supply is therefore optimized. Altogether, with the concomitant increase in arterial blood pressure, coronary perfusion pressure improves significantly along with the RV function. Finally, end-organ perfusion is maximized by the arterial retrograde flow associated with the lower central venous pressure.

***Is VA-ECMO associated with better survival in high-risk PE?*** Several retrospective studies were recently published and supported the use of VA-ECMO as a life-saving procedure in massive PE [9,11,37]. However, existing data mainly come from single-center studies with small sample sizes. Besides, the context (e.g., contraindication to systemic thrombolysis, failed thrombolysis, or failed alternative reperfusion strategies) and the severity (cardiogenic shock or cardiopulmonary resuscitation) at ECMO onset vary between patients and studies. In this context, the outcome-related data vary widely and are difficult to interpret.

***VA-ECMO and anticoagulation as a stand-alone therapy.*** Several authors hypothesize that VA-ECMO could be used as a stand-alone therapy until heparin-induced and spontaneous endogenous thrombolysis occur. Numerous retrospective studies have reported conflicting results with regard to this therapeutic option [9,11,37]. Maggio et al. published one of the first retrospective studies to promote this strategy [37]. Over 14 years (January 1992–December 2005), 21 patients received ECLS for high-risk PE. The mortality rate was 38% and neurologic injuries were the most common cause of mortality, accounting for 50% of the deaths. Interestingly, 76% of the survivors required no additional therapy other than VA-ECMO and anticoagulation. Significant clot dissolution occurred within 5 days, allowing VA-ECMO weaning. Similarly, Corsi et al. reported the experience of a tertiary-care center (Pitié-Salpêtrière Hospital in Paris, France) [11]. During the study period (2006–2015), 17 patients received ECMO support. Patients had a severe clinical presentation (median SAPS II of 78), 82% had cardiac arrest before ECMO support and 41% of them were cannulated during cardiopulmonary resuscitation. The 90-day mortality rate was 53%. Sixty-one percent of them received only ECMO and anticoagulation without additional reperfusion therapies, with ECMO removal after a median of 4 days. In a small case series, complete resolution or improvements in the clot burden on follow-up CT scan were reported with this strategy of ECMO and anticoagulation alone. Those findings were recently confirmed by Giraud et al. who reported a survival rate above 80% in 36 patients treated with VA-ECMO as a stand-alone strategy with a mean duration of ECMO support of 4 days in survivors [9]. When compared with other reperfusion strategies (thrombolysis, catheter-delivery thrombolysis), patients treated with an ECMO stand-alone strategy had a greater survival rate and experienced fewer bleeding complications.

However, larger retrospective studies reported conflicting results with this strategy [7,10]. In a multicenter series of ECMO-supported PE patients with refractory obstructive shock or cardiac arrest [7], the outcome was assessed according to the strategy used between thrombolysis and ECMO, ECMO and surgical embolectomy, or ECMO alone. The overall mortality of these 52 patients was 61.5%. The ECMO stand-alone approach was associated with worse outcomes. Indeed, the 30-day mortality was 76.5% in patients receiving ECMO and systemic fibrinolysis, 29.4% for those with ECMO and surgical embolectomy, and 77.7% in the ECMO alone group (*p* = 0.004). Based on these results, the authors concluded, “ECMO does not appear justified as a stand-alone treatment strategy in PE patients”. Another single-center retrospective study reported comparable mortality rates (69% in the ECMO alone group and 5% in the ECMO-embolectomy group) [10].

Nonetheless, these results should be interpreted with caution as ECLS was more frequently initiated during refractory cardiac arrest in the ECMO alone group. Selection bias was therefore a possible cofounder, as patients sent to surgery may have been in better clinical condition.

***VA-ECMO and surgical embolectomy.*** As stated above, VA-ECMO offers the advantage of stabilizing patients and it provides a bridge to additional perfusion therapy. A growing body of data suggests that surgical embolectomy is associated with excellent short- and long-term outcomes in high-risk PE patients. In this context, the combined approach of ECLS and surgery appears appealing although there is only limited data to support this combined strategy.

The largest study published so far, reported only 29% mortality in the group treated with ECMO and surgical embolectomy while other strategies were associated with dismal outcomes [7]. However, severe bleeding complications were more frequent in patients who received surgical thrombectomy after ECMO (54%). Several retrospective studies also confirmed this tendency [10,38].

Rather than opposing the ECMO stand-alone and the ECMO and surgical thrombectomy strategies, a stepwise algorithm that combines both strategies appears promising (Figure 1) [39,40]. In a high-volume surgical embolectomy center, the outcomes of 56 patients were analyzed according to the strategy chosen [39]. A “historical group” (27 patients) received surgical management as first-line therapy and VA-ECMO as salvage therapy. The second group of 29 patients was supported early by VA-ECMO and patients were bridged to either recovery or surgical thrombectomy after at least 5 days on ECMO. During the period on ECMO, anticoagulation was pursued, organ perfusion was ensured, and neurological function was assessed. If RV function improved within these 5 days, VA-ECMO was stopped and no additional reperfusion therapies were performed. While baseline characteristics did not differ across the groups, one-year survival was significantly lower in the “historical group” compared to the second strategy (73% vs. 96%; *p* = 0.02). Besides, several findings of this study illustrated the benefits of VA-ECMO in the combined strategy. First, it avoids the risk of pre-operative cardiac arrest. Indeed, 15% of the “historical group” presented cardiac arrest after general anesthesia. Second, postoperative cardiogenic shock was frequently reported in the first group, which is a known risk factor for multiple organ failure and an increase in postoperative morbidity. Third, neurological evaluation became possible before this high-risk procedure to avoid futile surgery in patients with potential preoperative irreversible brain injuries. Notably, 15% of the patients in the “historical group” presented severe anoxic injury, and care was withdrawn after a prolonged course of treatment. Lastly, this approach decreased the need for surgical intervention, which was performed in patients with persistent RV failure after 5 days of ECMO support. On the other hand, 52% of the patients in the second group recovered on ECLS alone. Although those results seem promising, the risk–benefit balance of this approach remains unknown and requires further validation.

***VA-ECMO and catheter-guided thrombolysis.*** Data regarding the use of catheter-directed thrombolysis associated with ECMO in PE patients are scarce (three studies with 36 patients). As with surgical embolectomy, pre-emptive VA-ECMO could be implemented in the presence of cardiogenic shock, to stabilize the patient and bridge them safely to that procedure. Georges et al. reported a mortality of 25% among 16 patients with combined procedures whereas it was 100% among patients with systemic thrombolysis only [51]. Further research is required to assess the effectiveness and safety of this strategy.

***VA-ECMO and long-term outcomes*****.** Few studies have investigated the long-term outcome of patients requiring ECMO for high-risk PE. Stadlbauer et al. analyzed the data of 119 high-risk PE patients supported by VA-ECMO of whom 67% had ECMO during or after cardiopulmonary resuscitation (CPR) [52]. The overall survival rate was 45.4%. At a median follow-up of 54 months, 34 patients were evaluated for long-term outcomes and quality of life. Cardiopulmonary function and exercise capacity were reported in 20 patients. Echocardiography did not reveal any signs of RV dysfunction or pulmonary hypertension whereas the pulmonary function test was only slightly decreased. No limitations in motor activity and mobility were reported in 73% of the patients and quality of life was slightly impaired compared to an age-matched reference population. Corsi et al. reported similar findings in seven patients, at a mean follow-up of 19 months [11]. Of note, only 28% returned to their previous work.

***E-CPR in PE: is it too late?*** As in many interventions, the timing of ECMO onset is of utmost importance. Indeed, survival rates were constantly significantly higher when ECMO was implanted for PE-associated cardiogenic/obstructive shock compared to during E-CPR [7,9,31,33,41,42,43,44,45,46,47,53,54,55] (Figure 1). For instance, Meneveau et al. observed a survival rate of 11% in high-risk PE patients undergoing E-CPR as compared to an overall survival rate of 52% for patients on ECMO for cardiogenic shock [7]. Similarly, the survival rate was only 9% in patients with PE on E-CPR whereas it was 42% when ECMO was initiated for refractory cardiogenic shock [45]. A recent meta-analysis confirmed that physicians tend to use ECMO as a rescue last-stage therapy, especially in PE [53,56]. Among 327 high-risk PE patients from 17 studies, ECMO was implanted for refractory cardiogenic shock in 140 (43%) patients and 187 (57%) patients for cardiac arrest [56]. Nonetheless, the mortality rate in PE-associated cardiac arrest without VA-ECMO is even higher and reaches 95% [4]. In such a context, what is the best strategy in the setting of PE-associated cardiac arrest? Should thrombolysis be used in such a situation or ECLS performed as a first-line therapy to minimize the low-flow time and improve the prognosis? Several authors have questioned the effectiveness of thrombolysis in PE-associated cardiac arrest and highlighted the severe complications related to this therapy. Giraud et al. reported a 20-fold increase in the rate of major bleeding when ECMO was implanted after systemic thrombolysis and an increase in 30-day mortality [9]. Several studies have also confirmed that pre-ECMO thrombolysis significantly increased the risk of bleeding complications [7,34,45].

A future RCT in the context of PE-related cardiac arrest is unlikely. Data from registries with a large sample size could offer a glimpse into this important clinical question. In this regard, Hobohm et al. investigated the outcomes of patients with PE deteriorating to cardiac arrest over 14 years [57]. Among 1,172,354 patients hospitalized with PE in Germany, of which 2197 had ECLS, the incidence of cardiac arrest was 6.5% (77,196 patients). Systemic thrombolysis was used in 27% of those patients (20,839) while a minority received thrombolysis and VA-ECMO (0.2%), embolectomy and VA-ECMO (0.5%), or VA-ECMO alone (0.8%). The highest mortality rate was observed among patients treated with systemic thrombolysis (83.3%). The mortality rate among patients supported by ECMO was 61.8%. With regard to cardiac arrest, multivariable logistic regression analysis showed a lower risk of in-hospital mortality in patients who were treated with VA-ECMO alone or in combination with other reperfusion strategies. This is in line with the data from several retrospective studies discussed previously [48,54].

Altogether, those results suggest that E-CPR in the context of massive PE seems to be associated with a higher survival rate compared to systemic thrombolysis. Its use should be discussed on a case-to-case basis, as pre-ECMO thrombolysis, especially in the context of cardiac arrest, appears to be associated with severe complications and worse outcomes. Additional investigations are required to clarify the optimal sequence and timing of therapies in this highly selected population.

***What are the latest recommendations?*** While the physiological rationale is appealing, the use of VA-ECMO in the setting of high-risk PE is not supported by any RCTs and remains poorly defined. In this context, the 2019 ESC guidelines stated “the use of VA-ECMO as a stand-alone technique with anticoagulation is controversial” and is “of no clear clinical benefit unless combined with surgical embolectomy”. Finally, the European guidelines recommend that ECMO “may be considered in combination with alternative reperfusion strategy, in patients with PE and refractory circulatory collapse or cardiac arrest” (Class IIb, Level C) [5].

***VA-ECMO and PE: key points for implantation.*** Several key practical points should be discussed regarding the management of PE patients requiring ECLS. In the context of cardiogenic shock and contraindication to thrombolysis, ECLS should be implanted under local anesthesia and procedural sedation in a spontaneously breathing patient. Indeed, mechanical ventilation in the context of acute RV failure should be avoided at all costs as an additional increase in the RV afterload may precipitate circulatory collapse. In addition, after failed thrombolysis, hemorrhagic complications must be anticipated accordingly when VA-ECMO is used as a rescue therapy. In centers where intensivists or cardiologists carry out the procedure, we believe that vascular or cardiac surgeons should be available during the cannulation to face potential catastrophic complications associated with canulation and thrombolysis-induced coagulopathy. A practical algorithm for ECMO-decision and care management of high-risk pulmonary embolism is proposed in Figure 2.

## 3. Research Agenda

More than 60 years after the first trial on systemic thrombolysis, the optimal management of high-risk PE remains controversial. The latest guidelines pointed out that thrombolysis is still the treatment with the highest level of evidence, which is factually correct. Nonetheless, the latest trial supporting this strategy was conducted in 1995 and only included eight patients. In addition, meta-analyses of RCTs supporting the efficacy of thrombolysis included only a minority of high-risk PE patients.

There is a growing body of evidence suggesting there is room for other strategies in the algorithm management of this population. VA-ECMO support appears suitable either to reverse the hemodynamic impairment and bridge patients to recovery or to further reperfusion therapies. Surgical embolectomy seems to be associated with a higher survival rate than catheter-guided thrombolysis, especially when combined with VA-ECMO. However, to date, there is no RCT comparing any of these strategies with systemic thrombolysis. It may be time to conduct this trial to optimize the treatment and prognosis of those patients. Beyond short-term mortality, the safety of these strategies should be carefully assessed.

As highlighted in this narrative review, one of the main limiting factors of such RCTs will be the sample size, as studies published from high-volume tertiary centers and often conducted over more than 10 years, rarely included more than 100 patients (Table 1). An international and multi-center trial conducted over several years appears mandatory to achieve these goals. 

## 4. Conclusions

High-risk PE is a life-threatening condition. Compared to other cardiovascular diseases, research has been limited and physicians have a limited therapeutic arsenal at their disposal. The main standard treatment is still systemic thrombolysis. However, the use of VA-ECMO has increased worldwide in the last decades. In case of thrombolysis contraindications or failure, using VA-ECMO as a stand-alone therapy or as a bridge to alternative reperfusion strategies seems wise and is associated with acceptable outcomes. Further studies, especially RCTs are urgently needed to better define the optimal care for this highly selected population.

## Figures and Tables

**Figure 1 jcm-11-04734-f001:**
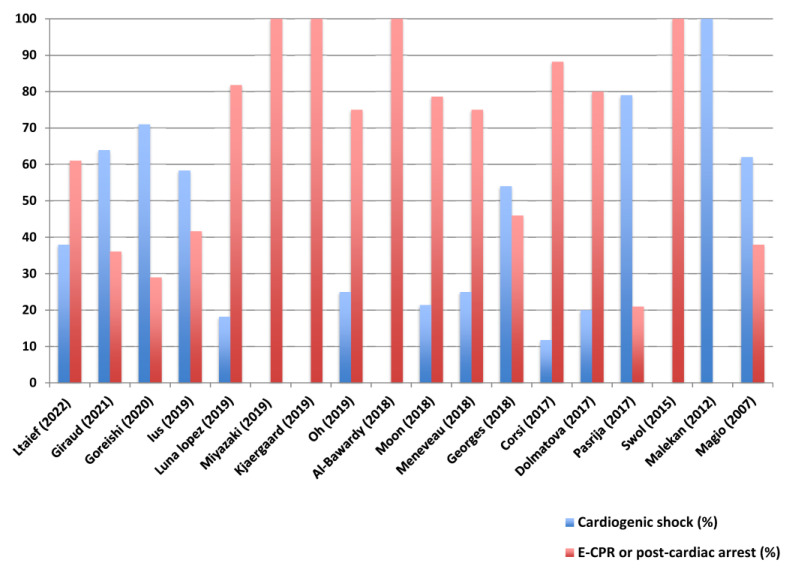
Percentage of patients cannulated during CPR/immediately after cardiac arrest or in cardiogenic shock in the main studies on VA-ECMO use in high-risk PE. E-CPR, Extracorporeal cardiopulmonary resuscitation; PE, pulmonary embolism; VA ECMO, venoarterial extracorporeal membrane oxygenation [7,9,10,11,19,37,39,40,41,42,43,44,45,46,47,48,49,50].

**Figure 2 jcm-11-04734-f002:**
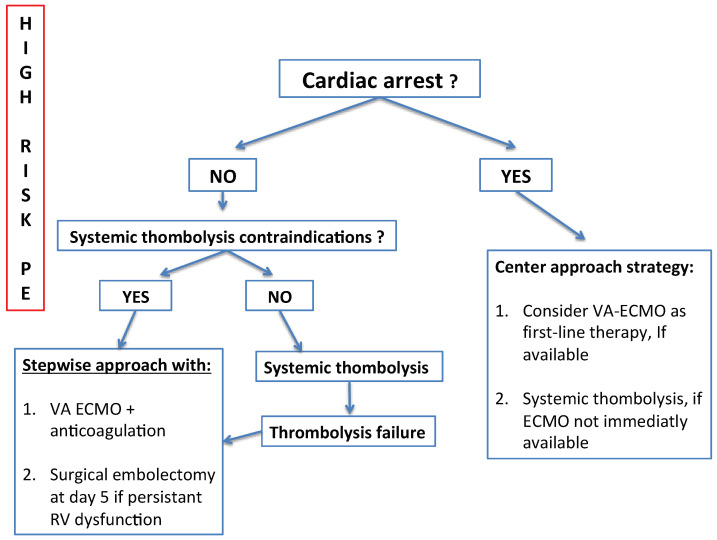
Proposal algorithm for care management of high-risk pulmonary embolism. RV, right ventricle; PE, pulmonary embolism; VA ECMO, venoarterial extracorporeal membrane oxygenation.

**Table 1 jcm-11-04734-t001:** Large and recent studies of ECMO for pulmonary embolism: key patient features.

Study	Study Period	Population (n)	Age (Years)	Cardiogenic Shock (%)	E-CPR/ Cardiac Arrest (%)	Systemic Thrombolyis Prior ECMO (%)	ECMO Alone (%)	CDT (%)	Surgical Embolectomy (%)	Overall Mortality (%)	Mean Ecmo Support (Days)
Ltaief (2022) [45]	2008–2020	20	57 (IQR 47–66)	38	61	50	44.4	0.5	22	88	NA
Giraud (2021) [9]	2010–2019	36	57 (IQR 23)	63.9	36.1	44.4	52	15.6	0	36	3.2 ± 3.2
Goreishi (2020) [40]	2015–2018	41	51 ± 15	71	29	21.9	51.3	2.4	24.4	NA	6.3 ± 2
Ius (2019) [10]	2012–2018	36	56 *	58.3	41.7	25	19.6	27.7	27.7	33	6
Luna lopez (2019) [49]	2013–2018	11	60 ± 8.5	18.2	81.8	53.8	27.2	38	0	54.5	3.5 ± 1.9
Miyazaki (2019) [41]	2014–2017	9	50 ± 16.1	0	100	44.4	11.1	55.5	11.1	11.1	2.8 ± 1.1
Kjaergaard (2019) [44]	2004–2017	36	55 ± 16.7	0	100	58.3	27.7	2.7	11.1	31	1.2 ± 1.8
Oh (2019) [43]	2014–2018	16	51 (IQR 38–70)	25	75	25	18.8	18.8	37.5	43.8	1.5 (IQR 0–4.5)
Al-Bawardy (2018) [47]	2012–2018	13	49 ±19	0	100	38.4	7.6	23	30.7	31	5.5
Moon (2018) [48]	2010–2017	14	53.6 ±17.7	21.4	78.6	7.1	85.7	0	7.1	64.3	8 ± 8.1
Meneveau (2018) [7]	2014–2015	52	47.6 ± 15	25	75	32.7	34.6	0	32.7	48.3	NA
Georges (2018) [19]	2012–2015	32	56 (IQR 46–66)	54	46	15.6	18.8	59.3	6.25	46.9	7.8 (IQR 1.7–11)
Corsi (2017) [11]	2006–2015	17	51 ±15.9	11.8	88.2	47	29.4	11.8	11.8	53	4 ± 3.4
Dolmatova (2017) [42]	2011–2015	5	52 ± 11.5	20	80	20	20	40	20	40	10.4 ± 4.4
Pasrija (2017) [39]	2014–2016	20	47 (IQR 32–59)	79	21	0	40	20	45	10.7	5.1 (IQR 3.7–6.7)
Swol (2015) [46]	2008–2014	5	45 ± 6.3	0	100	60	20	20	20	40	1.9 ± 1.7
Malekan (2012) [50]	2005–2011	4	46.8 ± 20	100	0	0	75	0	25	0	6.5 ± 2.3
Maggio (2007) [37]	1992–2005	21	41 *	62	38	28.5	76	0	19	38	5.4

* Results are expressed as a median. Patients could have received more than one reperfusion strategy in the included studies. Patients were cannulated either during cardiac arrest or immediately after. E-CPR, Extracorporeal cardiopulmonary resuscitation; ECMO, extracorporeal membrane oxygenation; CDT, catheter-delivery therapy; IQR, interquartile range; NA, non-applicable.

## Data Availability

Not Applicable.

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
