# Peer review of "Management of High-Risk Pulmonary Embolism: What Is the Place of Extracorporeal Membrane Oxygenation?"

_jcm, 2022, doi:10.3390/jcm11164734_

Round 1

Reviewer 1 Report

Main impression: this paper is well written and tackles a clinically relevant question for which hard evidence is lacking. I have found the narrative review format very appropriate to convey the most important messages in a way that makes the paper a pleasant read. I believe the paper deserves publication pending minor changes.

Major comment:

I believe that the core of the manuscript is between lines 207 and 290, and these paragraphs are well summarized in figure 1. The text reports quite exhaustively on current evidence from numerous papers. However, as stated by the authors, current evidence mostly stems from imperfect studies (limited sample size, single-centred, retrospective design) and have included patients with very different indications for ECMO and initial severity. Because the authors work in an expert centre and because they have adopted the design of a narrative review, I would thus encourage them to move away from merely summarizing the results of a list of studies, and more towards discussing/reporting their own experience and day-to-day practice when confronted to given clinical situations. Another way of saying this is that I think the paper would be even more interesting (to the non-expert reader) if the text between lines 207 and 290 looked more like figure 1.

Maybe one way of achieving this would be to discuss studies based on the initial indication for ECMO? I would be interested in knowing what is done at Pitie Salpetriere (and other centres?) in the following cases:

- Cardiac arrest: thrombolysis or first-line ECMO (and why?) à this is very appropriately discussed in lines 322-334.

- Failed systemic thrombolysis. This would lead to discuss major bleeding complications associated with ECMO implantation immediately after thrombolysis: should we avoid ECMO at all cost in those cases? What can be done to minimize bleeding risk?

- Contraindication to thrombolysis. This would lead to discuss how practically to implant ECMO, more specifically: under general anaesthesia or not? If current practice is to implant ECMO in sedated but not intubated patients, I doubt that this is regarded by all ICU physicians as standard practice, and I think this (key) point should be mentioned in one way or another in the text.

Specific comments/editing:

- Line 69: the comma in the middle of the sentence is probably not where it should be?

- Line 71: “population” not entirely clear, replace by “patients”?

- Line 130: add comma after “besides”?

- Line 134: “that rapidly restores”?

- Line 154 onwards: the discussion of reference 6 is not perfectly clear, would it be necessary to start with a short sentence explaining the setting and type of study?

- Line 180: I wouldn’t use “mechanical lysis” because this is then split into actual mechanical methods and also pharmacological + aspiration methods.

- Line 210: add comma after “besides”?

- Line 256: replace “a few” by “limited”?

- Line 293 + 302: is CPR defined before using abbreviation?

- Line 319: “increaseD”?

Author Response

Comment #1: This paper is well written and tackles a clinically relevant question for which hard evidence is lacking. I have found the narrative review format very appropriate to convey the most important messages in a way that makes the paper a pleasant read. I believe the paper deserves publication pending minor changes. 

Response: We thank reviewer 1 for the positive appreciation of our work.

Comment #2 : I believe that the core of the manuscript is between lines 207 and 290, and these paragraphs are well summarized in figure 1. The text reports quite exhaustively on current evidence from numerous papers. However, as stated by the authors, current evidence mostly stems from imperfect studies (limited sample size, single-centred, retrospective design) and have included patients with very different indications for ECMO and initial severity. Because the authors work in an expert centre and because they have adopted the design of a narrative review, I would thus encourage them to move away from merely summarizing the results of a list of studies, and more towards discussing/reporting their own experience and day-to-day practice when confronted to given clinical situations. Another way of saying this is that I think the paper would be even more interesting (to the non-expert reader) if the text between lines 207 and 290 looked more like figure 1. Maybe one way of achieving this would be to discuss studies based on the initial indication for ECMO? I would be interested in knowing what is done at Pitie Salpetriere (and other centres?) in the following cases:

- Cardiac arrest: thrombolysis or first-line ECMO (and why?) à this is very appropriately discussed in lines 322-334.

- Failed systemic thrombolysis. This would lead to discussing major bleeding complications associated with ECMO implantation immediately after thrombolysis: should we avoid ECMO at all cost in those cases? What can be done to minimize bleeding risk?

- Contraindication to thrombolysis. This would lead to discuss how practically to implant ECMO, more specifically: under general anesthesia or not? If current practice is to implant ECMO in sedated but not intubated patients, I doubt that this is regarded by all ICU physicians as standard practice, and I think this (key) point should be mentioned in one way or another in the text.

 Response: We thank the reviewer for this wise suggestion. Accordingly, we provide a new paragraph entitled VA-ECMO and PE: key points for implantation” in the discussion.

Comment #2 Specific comments/editing:

- Line 69: the comma in the middle of the sentence is probably not where it should be?

- Line 71: “population” not entirely clear, replace by “patients”?

- Line 130: add comma after “besides”?

- Line 134: “that rapidly restores”?

- Line 154 onwards: the discussion of reference 6 is not perfectly clear, would it be necessary to start with a short sentence explaining the setting and type of study?

- Line 180: I wouldn’t use “mechanical lysis” because this is then split into actual mechanical methods and also pharmacological + aspiration methods.

- Line 210: add comma after “besides”?

- Line 256: replace “a few” by “limited”?

- Line 293 + 302: is CPR defined before using abbreviation?

- Line 319: “increaseD”?

Response: We thank you for your comment and apologize for those typos. Changes have been done accordingly.

Reviewer 2 Report

I applaud the authors for this exceptional effort. 

In this narrative review the authors investigate and review the utilization of venoarterial extracorporeal membrane oxygenation in the management of high risk PE.  The authors note that the level of evidence supporting ECMO and alternative reperfusion therapy is low while the optimal management of high risk PE patients still evolving.

This Narrative review was evaluated on  the basis of the authors articulation of the articles importance for leadership, statement of specific aims/  questions  formulated, description of the literature search, scientific reasoning, appropriate  referencing and appropriate presentation of data.

Overall the article is very well written and no immediate concerns were observed

Introduction [60-81].  Justification of the narrative review is clearly  articulated.  Specific aims  and questions have been clearly  proposed at the  the end of the introduction

Overall the quality of the scientific appointment made and key arguments proposed have been done thoroughly with the presentation of concrete outcome data. Available level of evidence has been mentioned when able and available

I have but very few suggestion

Physiological rationale [197-206]-could benefit with references

Could the authors provide more insight into the use of VA ECMO as a preemptive therapy prior to catheter guided thrombolysis.  What group of patients of any could benefit from the same? 

Figure 1 -please redo not visible

Author Response

Reviewer 2

Comment #1. I applaud the authors for this exceptional effort. In this narrative review the authors investigate and review the utilization of venoarterial extracorporeal membrane oxygenation in the management of high risk PE.  The authors note that the level of evidence supporting ECMO and alternative reperfusion therapy is low while the optimal management of high risk PE patients still evolving. This Narrative review was evaluated on  the basis of the authors articulation of the articles importance for leadership, statement of specific aims/  questions  formulated, description of the literature search, scientific reasoning, appropriate  referencing and appropriate presentation of data.

Overall the article is very well written and no immediate concerns were observed. Introduction [60-81].  Justification of the narrative review is clearly  articulated.  Specific aims  and questions have been clearly  proposed at the  end of the introduction. Overall the quality of the scientific appointment made and key arguments proposed have been done thoroughly with the presentation of concrete outcome data. Available level of evidence has been mentioned when able and available

Response: We would like to sincerely thank the reviewers for her/his positive comments.

Comment #2. I have but very few suggestions. Physiological rationale [197-206]-could benefit with references.

Response: These references have been added.

Comment #3. Could the authors provide more insight into the use of VA ECMO as a preemptive therapy prior to catheter guided thrombolysis.  What group of patients of any could benefit from the same? 

Response: As requested, this point has been discussed as follows “As with surgical embolectomy, preemptive VA-ECMO could be implemented in the presence of cardiogenic shock, to stabilize the patient and bridge him safely to that procedure”.

Comment #4. Figure 1 -please redo not visible

Response: Done with appropriate modifications (now labelled Figure 2)

Reviewer 3 Report

In the current research article, " Management of high-risk pulmonary embolism: what is the place of extracorporeal membrane oxygenation? " authors argue about the use of VA-ECMO as a possible alternative for treatment of PE when traditional therapies such as systemic thrombolysis fail or are complicated to implement. The authors have discussed this issue very well. I liked the approach of starting with the pathophysiology of Pulmonary embolism. However, I think the diagrammatic representation will be useful for the readers for both PE and how VA-ECMO is implemented. But, overall I find this review very comprehensive.

Comment

Comment 1: Kindly include diagrammatic representations for PE and VA-ECMO.

Author Response

Reviewer 3

Comment #1. In the current research article, " Management of high-risk pulmonary embolism: what is the place of extracorporeal membrane oxygenation? " authors argue about the use of VA-ECMO as a possible alternative for treatment of PE when traditional therapies such as systemic thrombolysis fail or are complicated to implement. The authors have discussed this issue very well. I liked the approach of starting with the pathophysiology of Pulmonary embolism. However, I think the diagrammatic representation will be useful for the readers for both PE and how VA-ECMO is implemented. But, overall I find this review very comprehensive.

Response: We thank reviewer 3 for the positive appreciation of our work.

Comment #2. Kindly include diagrammatic representations for PE and VA-ECMO.

Response: We thank the reviewer for this interesting suggestion. We have added in Figure 1 the proportion of patients cannulated on CPR or in cardiogenic shock in the main studies on ECMO and PE.
